# Telehealth Interventions to Support Self-Management in Stroke Survivors: A Systematic Review

**DOI:** 10.3390/healthcare9040472

**Published:** 2021-04-15

**Authors:** Na-Kyoung Hwang, Ji-Su Park, Moon-Young Chang

**Affiliations:** 1Department of Occupational Therapy, Seoul North Municipal Hospital, 38 Yangwonyeokro, Jungnang-gu, Seoul 02062, Korea; occupation81@gmail.com; 2Advanced Human Resource Development Project Group for Health Care in Aging Friendly Industry, Dongseo University, 47 Jurye-ro, Sasang-gu, Busan 47011, Korea; jisu627@hanmail.net; 3Department of Occupational Therapy, Inje University, 197 Inje-ro, Gimhae 50834, Korea

**Keywords:** telehealth, self-management, stroke survivor, systematic review

## Abstract

Telehealth (TH) intervention is a method to optimize self-management (SM) support in stroke survivors. Objectives of this study included identifying the TH-SM intervention’s focus and SM support components, the TH delivery type, and the TH-SM support effects on stroke survivors. Five databases were searched for the years 2005–2020 to identify TH-SM support interventions for stroke survivors. Randomized controlled trials and quasi-experimental, one-group re-post study designs were included. Ten studies were reviewed. TH-SM support focused on post-stroke depression, obesity management, participation, functional mobility, and activities of daily living. The TH delivery type most used in selected studies was messaging. Regarding the SM support components, the education component was used in all studies, and psychological support and lifestyle advice and support were used in 8 out of 10 studies. TH-SM intervention had positive effects in terms of goal achievement for SM behavior, emotional state, and mobility of clinical outcomes, and TH acceptance in stroke survivors. Although the TH-SM-supported intervention effects were not found consistently in all outcomes, this review discovered a positive effect on various SM-related outcomes. In addition, TH delivery types and SM support components showed the possibility of various options to be considered for intervention. Therefore, we suggest that TH-SM supported intervention is a positive alternative for SM support in stroke survivors.

## 1. Introduction

In stroke care, emphasis has been placed on intense medical care and rehabilitation benefits in the acute phase, but care and support in the subsequent stages are as important as acute phase care [1]. People experiencing long-term health conditions, such as stroke survivors, must learn new behaviors or adjust their lifestyle to suit their needs [2].

Improving self-management (SM) is an important challenge for healthcare systems worldwide for long-term condition (LTC) survivorship [3]. According to the U.S. Institute of Medicine, SM is defined as tasks that one must perform while living with one or more chronic conditions [4]. The main elements are medical and behavioral management, emotional management, and role management, and the SM intervention empowers clients by providing them with the knowledge and skills to manage these tasks well [5]. For effective SM support implementation, Pearce et al. [6] developed the Practical Reviews in Self-Management Support (PRISMS) taxonomy, which consists of 14 components for SM support intervention in LTCs. Although many interventions use multiple components of SM support, the effective configuration and implementation of SM components is crucial [7]. Based on the PRISMS taxonomy, Hanlon et al. [8] proposed six components for SM support using telehealth (TH): patient education and information, monitoring using feedback and action plans, clinical reviews, adherence support, psychological support, and lifestyle interventions.

TH provides an appropriate alternative in healthcare, where the provision of acceptable and high-quality primary care is limited due to the increase in life expectancy and consequent chronic conditions [9]. It helps clients manage their condition through improved self-care and access to education and support systems. In addition, clients and health professionals can remotely exchange important clinical information for the management and support of LTCs [10]. Telerehabilitation has been conducted using telecommunication devices to provide evaluations and interventions for improving the motor, cognitive, and psychosocial function of stroke survivors [11,12]. It has become a means to meet the rehabilitation needs of stroke survivors living in rural areas and low- and middle-income countries, where stroke care is burdensome [13,14]. The increasing use of mobile phones and personal computers/tablets can be a valuable resource for lifestyle change and disease management. Specifically, the online eHealth support tool, accessible from a handheld device or personal computer, is one way to optimize SM support for clients’ goals [15,16].

With the background of various TH interventions and techniques, we aimed to obtain a broad overview of the evidence of TH-SM support in stroke survivors using a systematic review methodology. Therefore, the objectives of this review were to: (1) identify the focus of TH-SM intervention and SM support components in stroke survivors, (2) identify the type of TH delivery, and (3) identify the effects of the TH-SM support in stroke survivors.

## 2. Materials and Methods

### 2.1. Search Strategy and Selection Criteria

We searched five databases for reviews (Medline Complete, Embase, CINAHL, PsycINFO, and Web of Science) to identify relevant studies published on the TH-SM support interventions for stroke survivors between 2005 and 2020. Our basic search strategy included “telehealth terms” and “stroke terms” and “self-management support terms.” The following keywords were used: “telehealth,” “e-health,” “mobile health,” “telerehabilitation,” “telecare,” “telehealthcare,” and “stroke,” “cerebrovascular accident,” “CVA,” “cerebrovascular stroke,” “cerebrovascular apoplexy,” and “self-care,” “self-management,” “self-help,” “self-monitor,” “lifestyle,” “patient education.”

The eligibility criteria included: (1) individuals with a clinical diagnosis of stroke, (2) interventions focused on SM support, (3) studies using TH intervention, including telephone calls and/or web- or mobile-app-based interventions, and (4) studies with outcomes focused on goal attainment, self-care, activities of daily living, self-efficacy, quality of life, lifestyle behavior, participation, function, psychosocial, disability, and adherence. We excluded: (1) non-English publications, (2) single SM strategy interventions (a study focused on a specific strategy as opposed to interventions using broader strategies), (3) Studies involving only caregiver-related outcomes, and (4) dissertations, theses, and protocol studies. The study titles and abstracts were examined after initial search. Furthermore, we obtained the full text of eligible studies based on the eligibility and exclusion criteria. The manuscript was searched for eligibility and exclusion criteria. A consensus was sought between the authors for the study to be included in the systematic review. This review was guided by the Preferred Reporting Items for Systematic Reviews and Meta-Analyses (PRISMA) statement [17].

### 2.2. Quality Appraisal

Among the selected studies, the quality of randomized controlled trials (RCT) were appraised using the Physiotherapy Evidence Database (PEDro) Scale [18]. Three of the four studies showed “high” quality. The items with low scores on the scale used blind subjects and therapists (Table 1). The Risk of Bias Assessment tool for Non-randomized Studies (RoBANS) was used to appraise the quality of six non-randomized controlled trials (NRCT) [19]. In the comparability of participants, one study was evaluated as having a high risk of bias due to the pre- and post-design of the non-equivalent control group whose homogeneity between the intervention group and the control group was not verified. Four studies showed a high risk of bias in the confounding variables, as there was no group to control for confounding variables or a control group whose homogeneity was not verified. As none of the six studies mentioned evaluator blinding, the risk of bias was evaluated in terms of blinding the outcome assessment. Items other than these indicated a low risk of bias (Table 2).

### 2.3. Outcomes and Relevance

In this review, the following primary outcomes of the TH-SM intervention were targeted: (1) SM behaviors, goal achievement, activities of daily living (ADL), participation, and medication adherence; (2) clinical outcome; disability or recovery level, physical function, mobility, fatigue, and emotion; (3) self-efficacy; (4) quality of life (QOL); and (5) TH acceptance, adherence, subjective feedback, and satisfaction.

## 3. Results

Of the 1028 titles and abstracts, ten studies were included in this review of TH-SM interventions for stroke survivors [20,21,22,23,24,25,26,27,28]. Figure 1 shows a PRISMA flow chart shown in Figure 1 illustrates the search results and review selection. Two reviewers screened the full texts independently. The data were completed by one reviewer using a data extraction table, subsequently checked by a second reviewer for accuracy, with discrepancies resolved by discussion.

### 3.1. Participants’ Characteristics

A total of 427 participants were included in ten studies. Sample sizes varied throughout the studies, ranging from 12 [26] to 147 participants [24]. The phases of recovery after participants’ stroke was acute phase (*n* = 2), chronic phase (*n* = 1), and both acute and chronic survivors for 3–8 months or less than 24 months (*n* = 5). One study did not mention the time course (Table 3).

### 3.2. Types of TH delivery

Messaging was the most common in the selected studies, provided through short message service (SMS), email, in-home messaging devices (IHMDs), and smartphone push notification. All but one study [20] using messaging services combined with phone calls or web-based [29] or app-based interventions [23]. Four studies used phone calls [21,22,27,28], while four others used web-based intervention, such as video conferencing [24,25,26] or a web platform [24,29]. One study used an app platform for smart devices [23]. The TH types were synchronous (*n* = 2), asynchronous (*n* = 2), or both (*n* = 6) (Table 3).

### 3.3. Frequency, Duration, and Length of Intervention

The frequency of intervention varied according to the characteristics of the intervention sessions. Messaging was used at a frequency of once a day [20,21,22,29] or twice a day [28], or once a week [23]. Telephone sessions were once a week [27], twice a week [28], or five times throughout the course of the intervention [21,22]. The video conferencing session was performed twice a week [25,26], or four times during the intervention period [24]. Three studies reported the duration of each phone call or video conferencing session as 5–15 min [27], 20 min [24], and 2 h [25]. In some studies, the exact duration of the intervention was not mentioned [20,21,22,28,29]. The length of interventions ranged from 1 to 6 months (Table 3).

### 3.4. Intervention

The focus of the TH-SM interventions varied. Cadilhac et al. [20] and Kamoen et al. [24] focused on lifestyle behavior changes and disease management. The remaining studies explored the relief of depressive symptoms after stroke [27], physical activity and participation [25,26], functional mobility [21,22], obesity management [23], and ADL [28,29] (Table 3). Most studies excluded stroke survivors with severe aphasia and cognitive decline, but one study included participants with cognitive decline if they could participate in the intervention with the help of their caregiver. The majority of interventions were home-based, while other studies included delivery in a specific place in the area where the participants lived, prepared in advance by researchers. 

Various SM support components were used in selected studies, and we analyzed the regrouped components from the PRISMS taxonomy of SM support for LTCs. Table 4 lists the SM support components analyzed in this study. All studies used at least four of nine components.

#### 3.4.1. Education: Stroke Related Issues and Self-Management

Education components were used in all ten studies. They addressed personal recovery and prevention post-stroke; level of functional ability; post-stroke depression (PSD); risk factor management; and exploration and solutions to problems affecting target behavior. 

#### 3.4.2. Information: Sources of Social or Peer Support or Adaptive Equipment

Five studies provided information on community resources, help from family and friends, patient support groups, modifying the home environment, and new adaptive equipment or techniques as one of the SM support components.

#### 3.4.3. Remote Monitoring with Feedback and Action Plans

Seven studies addressed the components by phone calls, SMS, push notification, messaging devices, and online platforms. These include depressive symptoms, medication adherence, daily caloric intake, visual feedback on cardiovascular risk, verbal feedback on self-measurement, performance evaluation of targeted behavior, and discussion of future behavior formation. 

#### 3.4.4. Training/Rehearsal for Everyday Activities

Two studies consisted of exercise to improve physical function as one of the intervention sessions, two studies included the daily performance of targeted ADL tasks, and one study addressed daily calorie intake and exercise. 

#### 3.4.5. Clinical Review: Regular Follow-Up Reviews

In seven studies, the client’s condition and SM by the healthcare expert were confirmed and reviewed weekly or after an intervention session, or specific time period during intervention.

#### 3.4.6. Adherence Support

Seven of the ten studies included a logbook of participants’ performance and symptoms, a reminder message about the task to be performed, and a motivational or positive encouragement message to enhance behavior adherence.

#### 3.4.7. Psychological Support: Goal Setting, Action Planning, and Problem Solving Strategies

The psychological support component was used in nine studies. Although the studies did not include all detailed strategies for goal setting, action planning, and problem-solving, or did not clearly describe detailed strategies, they included at least one detailed strategy.

#### 3.4.8. Social Support: Peer Support, Peer Mentoring, and Group Socialization

In two studies, social conversation and peer support among participants living in different regions were conducted through video conferencing.

#### 3.4.9. Lifestyle Advice and Support: Practical Advice in Relation to Handling Life Stressors

Studies included support, discussion, or education for lifestyle changes related to physical activity, nutrition, relaxation, ADL performance, and healthy behavior, excluding one.

### 3.5. Outcome Measures

Table 5 and Table 6 show the outcomes of TH-SM interventions.

#### 3.5.1. SM Behaviors

In our review, goal achievement (*n* = 5), SM skills (*n* = 1), ADL (*n* = 1), participation (*n* = 5), and medication adherence (*n* = 1) were found to be the outcomes of SM behavior. In studies using goal achievement as an outcome [20,25,26,28,29], only one study showed statistically significant improvement in group comparative studies [28], and one group of pre-and post-test studies reported an improvement in the TH-SM group, although this was not significant [26,29]. The results of goal achievement in various areas, such as function, participation, and environment, are shown [20,25]. Five studies measured the outcome of participation, and two studies reported a positive effect of improving participation [26,28].

#### 3.5.2. Clinical Outcomes

The level of disability (*n* = 4), emotional state (*n* = 5), physical function (*n* = 1), mobility (*n* = 2), fatigue (*n* = 1), cardiovascular risk (*n* = 1), and body weight (*n* = 1) were measured as clinical outcomes. Regarding disability outcome, Chumbler et al. [21] showed greater improvement in the TH-SM group than in the control group, although there was no significant difference between the groups. Regarding emotional status, one study reported significant improvement [26] and another study showed improvement within the group [27]. In addition, TH-SM intervention had positive effects on other clinical outcomes, except fatigue and body weight [21,23,24,25,26,27,29].

#### 3.5.3. Self-Efficacy

The efficacy of balance ability (*n* = 1), fall-efficacy (*n* = 1), and ADL performance (*n* = 1) were measured as self-efficacy outcomes. The TH-SM intervention had significant effects on all self-efficacy outcomes.

#### 3.5.4. QOL

QOL was measured in three studies, and only one study reported a significant improvement [24].

#### 3.5.5. TH Acceptance

Adherence (*n* = 4), subjective feedback (*n* = 6), acceptability (*n* = 1), and satisfaction (*n* = 1) were measured. Adherence was measured using participation or response rates. Acceptability and satisfaction were measured on a self-measured scale. Subjective feedback findings were collected through interviews with some participants or focus groups. All outcomes of intervention acceptance showed positive outcomes [20,22,23,24,25,26,27,29].

## 4. Discussion

### 4.1. Interventions

The major focus of TH-SM support in the selected studies was the relief of depressive symptoms, physical activity and participation, functional mobility, ADL, and obesity management in stroke survivors. PSD occurs in one-third of stroke survivors [30] and has a negative effect on the restriction of participation in the rehabilitation process, decline in physical, cognitive, and social functions, and the biological process of neuroplasticity [31]. However, according to previous studies, many patients with PSD are not treated, and accordingly, innovative strategies to identify and treat PSD are needed [32]. One study in this review used TH-SM support as one of the approaches to PSD and demonstrated positive results [27]. Participation restrictions are the difficulties encountered during the integration of stroke survivors into premorbid life roles [33]. The factors that determine the participation of stroke survivors are not only age and sex but also the survivor’s functional/physical ability, independence in ADL, the severity of the stroke, and the onset of depression [34]. Traditionally, most SM interventions have primarily evaluated clinical outcomes, and SM support has focused on medical and emotional management skills, such as proper medication adherence and stress management in chronic conditions. However, outcomes related to function or participation are also important indicators and goals of SM support that help to improve the client’s role management ability [35]. One study in this review addressed self-monitoring for obesity management [23]. Adherence to evidence-based approaches for obesity management in stroke patients is insufficient, but its effectiveness has been demonstrated in some limited healthcare interventions [36,37]. Recently smartphone-based self-monitoring intervention for obesity management has been documented as one of the useful mHealth interventions for managing stroke risk factors in stroke survivors with physical and cognitive challenges [38]. The studies included in this review provided TH-SM support, which focused on improving the role management ability and medical and emotional management skills of stroke survivors. It could be considered a necessary aspect for constructing SM support interventions with stroke survivors.

### 4.2. Types of TH Delivery

The nine SM support components of stroke survivors regrouped based on the PRISMS taxonomy for LTCs were provided through TH. Methods included messaging, telephone, video conferencing, and online platforms. In addition, TH intervention was provided either alone or in a combination of two types. Six out of ten studies used messaging as a tool to support, monitor, and provide customized information for target behavior. Champion et al. [39] reported that information on issues related to health behavior motivation and potential risk provided by messaging interventions could increase client knowledge and thus reduce health threats. This is consistent with the SM support interventions that can be loaded into the messaging delivery type identified in this review. Tele-communication can be adopted to improve the management of LTCs [40,41], and direct interactions, such as voice calls, between clients and healthcare professionals can contribute to building confidence with providing information and receiving immediate feedback [42]. The selected studies in this review used bidirectional telecommunication with phone calls or video conferencing, which were delivered for the purpose of monitoring psychological symptoms, medication adherence, and SM support education. Three studies provided information about stroke, cardiovascular risk and obesity management control monitoring, daily ratings on the client’s own performance, and communication with healthcare professionals using a digital platform [23,24,29]. Evidence-based interventions, along with appropriate levels of guidance through a digital platform, can promote disease-specific health behavior changes and help effectively manage a variety of users and conditions [43]. The different types of TH delivery identified in this review are effective means for supporting TH-SM in stroke survivors. This may help with selecting an appropriate TH delivery type that can be equipped with a client-specific SM support strategy.

### 4.3. Outcomes and Effects of TH-SM Support Intervention

The outcomes included in this review were SM behavior, clinical outcome, self-efficacy, QOL, and TH acceptance. We could not find a consistent improvement in the detailed outcomes. Nevertheless, the outcomes that showed positive effects were the goal achievement of SM behavior, emotional state and mobility of clinical outcomes, and TH acceptance. Goal achievement evaluations were used to facilitate goal setting and the program outcome evaluation tool. Goal setting is one of the core elements of SM support, especially in the rehabilitation setting, which is an important step in helping to promote community transitions among stroke survivors [44]. Mood and emotional disturbances are common symptoms of stroke survivors [45]. Three of the five studies that included the emotional state outcome reported the positive effects of TH-SM-supported interventions. Provided interventions were tailored suggestions for depressive symptoms, medication adherence, and obesity management monitoring, general SM education and exercise, and client-centered daily ADL tasks. Studies measuring mobility outcomes have provided physical exercise through video conferencing, and the results also showed positive effects on balance and walking ability. For stroke survivors who experience mobility restrictions after onset, appropriate physical activity is necessary for lifestyle changes and adaptation after acute care. Home telehealth can be effective for assessing the health care needs of stroke survivors and caregivers, as well as providing information and emotional support to them [46]. The TH acceptance addressed the participation rate and satisfaction survey of the provided SM-supported intervention and included the subjective opinions of the participants. Some subjective opinions on the technical deficiencies of TH are included in the provided intervention, but the TH acceptance of the participants was generally positive. Previous studies have reported remarkable acceptance and a positive attitude toward mobile-based intervention for stroke management [47]. However, the development and application of techniques that take into account the familiarity and comfort of stroke survivors and caregivers could be considered in the future.

### 4.4. Limitations

There have not been many TH-SM-supported studies in stroke survivors; therefore, only a few studies were included in this review. The focus of intervention varied for each method, so there was a limit to confirming consistent improvement in outcomes. In addition, the sample sizes were small, studies without a control group were included, and studies that included a control, waitlist, or usual care group did not provide in-person SM support intervention, which could not be compared with TM-SM-supported interventions. Stroke is an LTC, and it is a disease that requires changes in healthy behavior and continuous adaptation after onset. Therefore, it is necessary to confirm the long-term effects of SM intervention.

## 5. Conclusions

Although the TH-SM-supported intervention effects were not consistently found in all outcomes, this review discovered a positive effect on various SM-related outcomes. Therefore, we suggest TH-SM-supported intervention as an alternative method for SM support in stroke survivors. In addition, TH delivery types for SM support, the focus of interventions in stroke survivors, and the components or strategies used for effective implementation, which presents a variety of options for TH-SM support interventions and its applicability in stroke survivors. Based on these findings, various trials are needed to establish a consistent basis for the effectiveness of TH-SM intervention in stroke survivors in the future.

## Figures and Tables

**Figure 1 healthcare-09-00472-f001:**
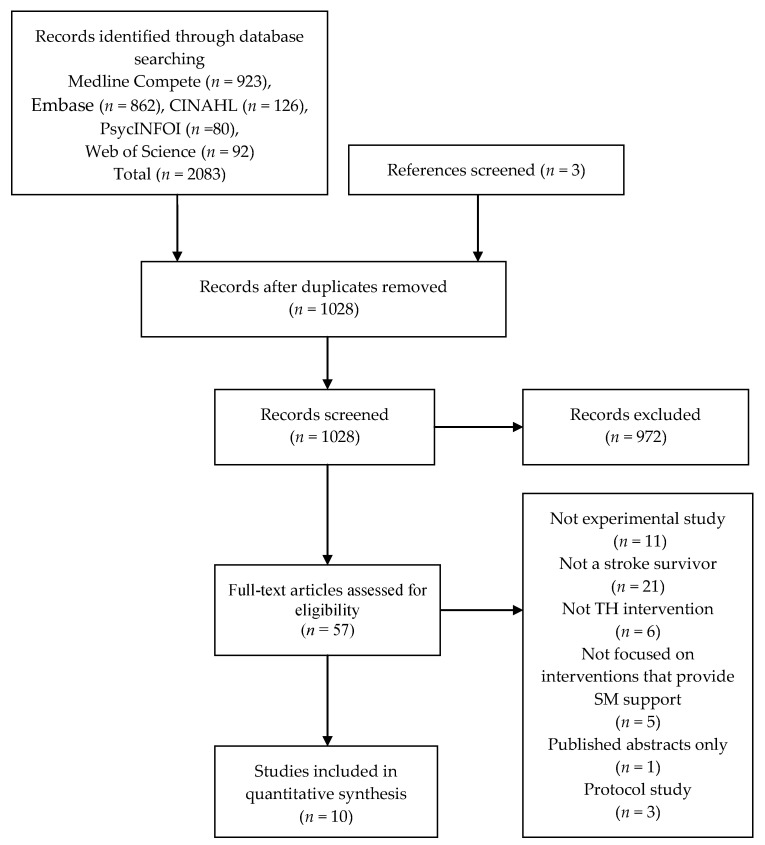
The Preferred Reporting Items for Systematic Reviews and Meta-Analyses (PRISMA) flow chart.

**Table 1 healthcare-09-00472-t001:** Summarized results of the quality assessment for RCT literatures using PEDro scale.

Autor, Year	Cadilhac et al., 2020 [20]	Chumbler et al., 2012 [21]	Chumbler et al., 2015 [22]	Ifejika et al., 2020 [23]
Eligibility	yes	yes	yes	yes
Random allocation	yes	yes	yes	yes
Concealed allocation	no	yes	yes	no
Baseline comparability	yes	yes	yes	yes
Blind subjects	no	no	no	no
Blind therapists	no	no	no	no
Blind assessors	yes	yes	yes	no
Adequate follow-up	yes	yes	yes	yes
Intention-to-treat analysis	yes	yes	yes	no
Between-group comparisons	yes	yes	yes	yes
Point estimated variability	yes	yes	yes	yes
Score; Quality	7/10; high	8/10; high	8/10; high	5/10; Fair

RCT: Randomized Controlled Trial, PEDro Scale: Physiotherapy Evidence Database Scale.

**Table 2 healthcare-09-00472-t002:** Summarized results of the quality evaluation for NRCT literature using RoBANS.

Autor, Year	Kamoen et al., 2019 [24]	Huijbregts et al., 2009 [25]	Taylor et al., 2009 [26]	Skolarus et al., 2019 [27]	Kamwesiga et al., 2018 [28]	Guidetti et al., 2020 [29]
Comparability of participants	low	low	low	low	high	low
Selection of participants	low	low	low	low	low	low
Confounding variables	low	low	high	high	high	high
Exposure measurement	low	low	low	low	low	low
Blinding of outcome assessment	high	high	high	high	high	high
Outcome assessment	low	low	low	low	low	low
Incomplete outcome data	low	low	low	low	low	low
Selective outcome reporting	low	low	low	low	low	low

NRCT: Non-Randomized Controlled Trial, RoBANS: Risk of Bias Assessment tool for Non-randomized Study.

**Table 3 healthcare-09-00472-t003:** Summary of studies investigating the use of TH-SM support intervention.

Author (Year)	Design; Participants	Time Post-Stroke	Telehealth Technology/ TH Type	Contents		Regime
				IG	CG	
Cadilhac et al., 2020 [20]	RCT IG = 29 CG = 25	Chronic stroke	Messaging Sync + Async	Personalized eHealth messages via SMS/email - Daily support messages matched to personal recovery and prevention goals and level of functional ability 2-way communication as needed 1–2 administrative message per week	Usual care Goal-setting assistance for 2–3 goals 2–3 administrative messages	4 weeks Up to 1 a day
Chumbler et al., 2012 [21]	RCT IG = 27 CG = 25	Acute- Chronic stroke survivors	Telephone & Messaging Sync + Async	Telephone-delivered intervention - Exploration of potential barriers and identification solutions - Instructions for the exercises and adaptive strategies In-home messaging device - Participant’s self-report measurement - Instant feedback with positive encouragement for exercise adherence	Usual care Home health care	3 months 3 times televisits 5 times telephone calls Message once a day
Chumbler et al., 2015 [22]	RCT IG = 27 CG = 25	Acute- Chronic stroke survivors	Telephone & Messaging Sync + Async	Telephone-delivered intervention - Exploration of potential barriers and identification solutions - Instructions for the exercises and adaptive strategies In-home messaging device - Participant’s self-report measurement - Instant feedback with positive encouragement for exercise adherence	Usual care Home health care	3 months 3 times televisits 5 times telephone calls Message once a day
Ifejika et al., 2020 [23]	RCT IG = 17 CG = 19	Acute stroke	App-based & Messaging Async	Smartphone-based weight loss self-monitoring intervention - Daily caloric intake monitoring, with reminder messages - 10% weight loss goal setting by researchers - Food information and nutrition data for achieving weight loss goals In-person visits: counseling, educational materials	Food journal self-monitoring Pocket-sized journal: calorie recording, food reference In-person visits	3 months Weekly push notification summaries of compliance
Kamoen et al., 2019 [24]	Nonequivalent Control Group IG = 94 CG = 53 uncontrolled	Acute stroke survivors	Web-based Sync + Async	Education during hospitalization: risk factor management, review of medication, clinical course and follow-up after hospitalization Video consultations after discharge: assessment of the stroke related problems Web platform - Displayed patient-specific neurological symptoms and cardiovascular risk factors, tips and tricks concerning a healthy lifestyle, patient support groups, useful apps	-	6 months 20 min educational session 20 min video consultations at 2 weeks, 1 month, 2 months and 6 months 1 message per video consultation (4 times in total)
Huijbregts et al., 2009 [25]	Nonequivalent Control Group IG = 10 CG = 8	Not stated	Web-based Sync	Video conferencing - Discussion session: stroke-related issues, problem-solving, and goal-setting skills - Exercise session: land-based exercise	Waiting list	9 weeks 2 sessions per week 2 h per session: 1 h of discussion, 1 h of exercise
Taylor et al., 2009 [26]	One group pre-post IG = 12	Acute- Chronic stroke survivors	Web-based Sync	Video conferencing - Discussion session: stroke-related issues, problem-solving, and goal-setting skills - Exercise session: warm-up, cardiovascular, balance and strength, cool down		9 weeks 2 sessions per week 2 h per session: 1 h of discussion, 1 h of exercise
Skolarus et al., 2019 [27]	One group pre-post Total = 13	Acute stroke survivors with at least moderate depressive symptoms	Telephone Async	IVR calls - Monitored both depressive symptoms and medication adherence along with tailored suggestions - Weekly IVR assessments Information sheets detailing the program + log books for tracking subjects’ symptoms + educational materials about depression	-	3 months weekly calls 5–15 min call
Kamwesiga et al., 2018 [28]	Nonequivalent Control Group IG = 15 CG = 15	Acute- Chronic stroke survivors	Telephone & Messaging Sync + Async	Mobile phone message intervention - Morning message: to remind the participant to perform three target activities during the day - Evening message: to respond daily performance scores, one for each activity Mobile phone calls from OT - Follow-up strategy guidance to explore and resolve issues related to goal achievement - Discussion and evaluation of the strategies implemented and formulation a new target with the client	Usual care	8 weeks Twice a day SMS Twice a week phone calls
Guidetti et al., 2020 [29]	One group pre-post IG = 13	Acute- Chronic stroke survivors	Web-based & Messaging Sync + Async	Web platform for person-centered approach - Viewed the daily alerts regarding the goals and strategies by the researcher(each morning) on web platform - Response with daily rating on web platform and logbooks during the day SMS daily alerts	-	8 weeks Once a day SMS alerts

TH: telehealth, SM: self-management, RCT: randomized controlled trials, IG: intervention group, CG: control group, Sync: synchronous, Async: asynchronous, IVR: interactive voice response, SMS: short message service, OT: occupational therapist, ICT: Information and Communications Technology.

**Table 4 healthcare-09-00472-t004:** SM support components.

Strategy	Cadilhac et al., 2020 [20]	Chumbler et al., 2012 [21]	Chumbler et al., 2015 [22]	Ifejika et al., 2020 [23]	Kamoen et al., 2019 [24]	Huijbregts et al., 2009 [25]	Taylor et al., 2009 [26]	Skolarus et al., 2019 [27]	Kamwesiga et al., 2018 [28]	Guidetti et al., 2020 [29]
Education: stroke related issues and SM	✓	✓	✓	✓	✓	✓	✓	✓	✓	✓
Information: sources of social or peer support or adaptive equipment	✕	✓	✓	✕	✓	✓	✓	✕	✕	✕
Remote monitoring with feedback and action plans	✕	✓	✓	✓	✓	✕	✕	✓	✓	✓
Training/rehearsal for everyday activities	✕	✕	✕	✓	✕	✓	✓	✕	✓	✓
Clinical review: regular follow-up reviews	✕	✓	✓	✓	✓	✕	✕	✓	✓	✓
Adherence support	✓	✓	✓	✓	✕	✕	✕	✓	✓	✓
Psychological support: goal setting, action planning, and problem solving strategies	✓	✓	✓	✓	✕	✓	✓	✓	✓	✓
Social support: peer support, peer mentoring, and group socialization	✕	✕	✕	✕	✕	✓	✓	✕	✕	✕
Lifestyle advice and support: practical advice in relation to handling life stressors	✓	✓	✓	✕	✓	✓	✓	✕	✓	✓
Number of components used	4/9	7/9	7/9	6/9	5/9	6/9	6/9	5/9	7/9	7/9

✓: component present, ✕: component absent/ unclear/ not specified. SM: self-management.

**Table 5 healthcare-09-00472-t005:** Summary of results of the included studies.

Author (Year)	Outcome Measures				Aim; Results
	Pre	Post		Assessment	
		T1	T2		
Cadilhac et al., 2020 [20]	BL	4 weeks		Goal achievement: GAS SM: heiQ Emotional status: HADS Participation: NEADL QOL: EQ-5D-3L	To assess the feasibility, acceptability and potential effectiveness of eHealth support messaging system; Achieved goal attainment (mean GAS-T score ≥ 50) related to function, participation and environment in the IG (CG: environment only) Non-significant differences between the groups for most SM domains and several QOL domains; potential improvements for SM and QOL domains in the IG compared with the CG Positive feedback and reports on eHealth messages: easy to understand (92%), helped achieve the goal (77%) in the IG
Chumbler et al., 2012 [21]	BL	3 months	6 months	Physical function: motor FONEFIM Function and disability: LLFDI	To determine the effect of stroke telerehabilitation on physical function and disability; Improvements of motor FONEFIM, LLFDI in the IG at 6 months; no significant difference between the groups Significant improvements in 4 of the 5 LLFDI disability subscales (*p* < 0.05), and approached significance in 1 of the 3 function subscales (*p* = 0.06) in the IG at 6 months
Chumbler et al., 2015 [22]	BL	3 months	6 months	Fall-related self-efficacy: FES Satisfaction with care: SSPSC	To determine the effect of stroke telerehabilitation in-home intervention on falls-related self-efficacy and patient satisfaction; Improvements of FES score in the IG than the CG; no significant difference between the groups Significant improvements of SSPSC in the IG Focus group interview - Reports from participants: exercises helpful, challenges using the in-home messaging device
Ifejika et al., 2020 [23]	BL	1 months	3 months/6 months (T3)	Body weight Depressive symptom: PHQ-9 Adherence: self-monitoring once daily for diet entry	To determine the feasibility and preliminary treatment effects of a smartphone-based weight loss intervention to monitor dietary patterns; No significant differences in weight loss between the IG and the CG (*p* = 0.77) Significantly lower PHQ-9 score at 1 month in the IG than in CG (*p* = 0.03); remained in the zero-minimal range for the IG compared with mild-moderate range in the CG at 3 and 6 months No significant differences in adherence between the groups
Kamoen et al., 2019 [24]	BL	6 months		Cardiovascular risk: SCORE Functional status and disability: mRS QOL: EQ-5D-5L Medication adherence	To test personal digital coaching program to improve cardiovascular risk factor control; Statistically significant reduction of SCORE (*p* < 0.001) in the IC No significant difference in SCORE between the IG and the CG Medication adherence of 96% in the IG Improved QOL quality of life (*p* < 0.001) in the IG No significant improvement in mRS in the IG Reports from participants: willingness recommend to others (96%), the impact on health literacy (86%)
Huijbregts et al., 2009 [25]	BL	9 weeks	18 weeks	Participation: RNL Well-being: SA-SIP 30 Mobility: BBS, CMSA-AI Goal achievement: GAS	To investigate the efficacy of telehealth delivery of SM program in improving aspects of community reintegration and well-being in community-dwelling persons with stroke; Significant difference in BBS between the IG and the CG (mean difference−4.27, 95% CI: −6.66 to−1.87) No significant differences in RNA, SA-SIP 30, CMSA-AI between the groups Improved GAS in the IG compared to the CG: primarily focused on physical activities and social participation Attendance and feasibility - Attendance rates for persons with stroke (83.9%), and care partners (76.7%) Focus group interview - Reports from participants: additional benefits including increased motivation and awareness of partners’ needs, decrease their sense of isolation
Taylor et al., 2009 [26]	BL	9 weeks	21 weeks	Goal achievement: LTG, STG Participation: RNL Emotional status: GDS Mobility: BBS, 6-MWT Balance confidence: ABC	To explore the feasibility of videoconference delivery of SM program to rural communities; Pre–post improvements were seen in goal setting, mood, balance, balance confidence, and walking endurance LTGs achievement 66%, weekly STGs achievement 68% Pre–post improvements in GDS, 6-MWT, ABC; significant difference in GDS, 6-MWT for post-hoc Focus group interview - Reports from stroke people and caregivers: greater awareness of stroke, increased social support, and improved ability to cope - Reports from and caregivers: motivated, learning to cope with change
Skolarus et al., 2016 [27]	BL	3 months		Depressive symptom: PHQ-9	To assess the feasibility and acceptability of IVR as an adjunct to post-stroke depression follow-up care; Improved PHQ-9 scores from a median score of 11 (IQR 7–13) at baseline to a mean of 4 (IQR 1–7, *p* = 0.11) at follow-up Reports from participants: good or excellent quality program, willingness recommend to others
Kamwesiga et al., 2018 [28]	BL	9 weeks		Goal achievement: COPM Self-efficacy in performance daily activities (developed by researchers) Perceived impact of stroke: SIS ADL: BI Participation: OGQ	To evaluate the feasibility of the mobile phone supported family-centered intervention, and the effects of the intervention; Significant difference in COPM performance component and self-efficacy between the IG and the CG Higher number of participants in IG with a 15-point clinically meaningful improvement in 6 of the 8 SIS domains Improvements of BI, OGQ in both group; no significant differences between the IG and the CG
Guidetti et al., 2020 [29]	BL	4 weeks	8 weeks	Goal achievement: COPM Perceived impact of stroke: SIS Frequency of participation: FAI Self-efficacy in performance daily activities: developed questionnaire by researchers Emotional status: HAD Fatigue: FSS Adherence: response rate Acceptability: open-ended questions	To evaluate the feasibility of (i) web-based family-centered intervention within in-patient and primary care rehabilitation after stroke, (ii) the study design and outcome measures used, and (iii) the fidelity, adherence, and acceptability of the intervention; Clinically meaningful improvement of ≥2 points of COPM: 4 participants regarding performance, 6 participants regarding satisfaction Improvement in different areas of SIS for each participant improvement of confident in performance ADL at both 4 weeks and at follow-up No changes regarding HAD, FSS Response rate to received messages: 78% Acceptability: positive feedback from all participants

BL: baseline, IG: intervention group, CG: control group, SM: self-management, GAS: Goal Attainment Scaling, heiQ: Health Education Impact Questionnaire, HADS: Hospital Anxiety and Depression Scale, NEADL: Nottingham Extended Activities of Daily Living, QOL: quality of life, EQ-5D-3L: EuroQoL-5dimension-3, IVR: interactive voice response, PHQ-9: Patient Health Questionnaire, mRS: modified Rankin scale, SCORE: Systematic COronary Risk Evaluation, RNL: Reintegration to Normal Living Index, SA-SIP 30: Stroke-Adapted Sickness Impact Profile, BBS: Berg Balance Scale, CMSA-AI: Chedoke-McMaster Stroke Assessment Activity Inventory, GDS: Geriatric Depression Scale, 6-MWT: 6-Minute Walk Test, ABC: Activity-Specific Balance Confidence Scale, LTG: long-term goal, STG: short-term goal, FES: Falls Efficacy Scale, SSPSC: Stroke-Specific Patient Satisfaction with Care, FONEFIM: Telephone Version of the Functional Independence Measure, LLFDI: Late-Life Function and Disability Instrument, COPM: Canadian Occupational Performance Measure, SIS: Stroke Impact Scale, ADL: activities daily of living, BI: Barthel Index, OGQ: Occupational Gaps Questionnaire, FAI: Frenchay Activities Index, FSS: Fatigue Severity Scale.

**Table 6 healthcare-09-00472-t006:** Effects of TH-SM support intervention.

Outcome		Number of Studies	RCT		NRCT			
					Nonequivalent Control Group		One-Group Pretest-Posttest	
			Study	Effect	Study	Effect	Study	Effect
SM behaviors	Goal attainment	5	Cadilhac et al., 2020 [20]	–	Huijbregts et al., 2009 [25] Kamwesiga et al., 2018 [28]	- +++(COPM; performance)	Taylor et al., 2009 [26] Guidetti et al., 2020 [29]	^ ^
	SM skill	1	Cadilhac et al., 2020 [20]	–				
	ADL	1			Kamwesiga et al., 2018 [28]	+		
	Participation	5	Cadilhac et al., 2020 [20]	-	Huijbregts et al., 2009 [25] Kamwesiga et al., 2018 [28]	- +	Taylor et al., 2009 [26] Guidetti et al., 2020 [29]	^ x
	Medication adherence	1			Kamoen et al., 2019 [24]	#		
Clinical outcome	Levels of disability	4	Chumbler et al., 2012 [21]	++	Kamoen et al., 2019 [24] Kamwesiga et al., 2018 [28]	- -	Guidetti et al., 2020 [29]	^(some sub-items)
	Emotional status	5	Cadilhac et al., 2020 [20] Ifejika et al., 2020 [23]	- +++			Skolarus et al., 2019 [27] Taylor et al., 2009 [26] Guidetti et al., 2020 [29]	^ +++ -
	Physical function	1	Chumbler et al., 2012 [21]	++				
	Mobility	2			Huijbregtset al., 2009 [25]	+++(BBS) -(CMSA-AI)	Taylor et al., 2009 [26]	+++(6MVT) ^(BBS)
	Fatigue	1					Guidetti et al., 2020 [29]	-
	Cardiovascular risk	1			Kamoen et al., 2019 [24]	+		
	Body weight	1	Ifejika et al., 2020 [23]	-				
Self-efficacy	Balancing ability	1					Taylor et al., 2009 [26]	+++
	Fall-related	1	Chumbler et al., 2015 [22]	++				
	Performance	2			Kamwesiga et al., 2018 [28]	+++	Guidetti et al., 2020 [29]	^
QOL		3	Cadilhac et al., 2020 [20]	-	Kamoen et al., 2019 [24] Huijbregts et al., 2009 [25]	+++ -		
TH acceptance	Adherence	4	Ifejika et al., 2020 [23]	+	Huijbregts et al., 2009 [25] Kamoen et al., 2019 [24]	# #	Guidetti et al., 2020 [29]	#
	Acceptability	1					Guidetti et al., 2020 [29]	#
	Satisfaction	1	Chumbler et al., 2015 [22]	+++(some sub-items)				
	Feedback from participant	6	Cadilhac et al., 2020 [20] Chumbler et al., 2015 [22]	# #	Kamoen et al., 2019 [24] Huijbregts et al., 2009 [25]	# #	Skolarus et al., 2019 [27] Taylor et al., 2009 [26]	# #

+++: statistically significant effect; ++: greater improvement in intervention group than control but between group difference not significant; +: significant improvement in both groups but between group difference not reported or not significant; -: no reported change in the group(s) or between the groups; x: effect-related data not shown; ^: within-group improvement not significant; #: consequences, such as high compliance or positive feedback, TH: telehealth, SM: self-management, RCT: Randomized Controlled Trial, NRCT: Non-Randomized Controlled Trial, QOL: quality of life.

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
