# Peer review of "Telehealth Interventions to Support Self-Management in Stroke Survivors: A Systematic Review"

_healthcare, 2021, doi:10.3390/healthcare9040472_

Round 1

Reviewer 1 Report

The paper addresses a topic that is relevant, however, probably only for a selected audience. The authors provide a good overview of the possibilities of long-term treatment for stroke patients. In the introduction, the authors emphasize the different dimensions of the problem, on an individual (from the patient's perspective) as well as on a societal level (healthcare systems).  In section 2, Materials and Methods, the authors describe their approach to the problem - a literature analysis on related work. The different steps and related decisions whether to include a publication or not in further analyses are clearly presented (and illustrated in Figure 1). Section 3 - Results - presents the contents of those studies, which were selected for further analyses. The authors point out the commonalities and differences, compare the studies on the basis of different characteristics (e.g. subject number) and draw conclusions. The paper finishes with a discussion and a conclusion. 

The topic of the paper is of interest, and the work / the efforts the authors have invested in are well documented. However, there are some weaknesses in the presentation. Specifically, the tables (3, 5, 6) are difficult to read and understand.  In some of the cases, tables cover more than one page, but labels are not shown on all pages. In regard to the tables' contents, I think that the details of selected studies could have been condensed to a higher extend. Some information is given repeatedly in each cell (e.g. table 3, telephone intervention, web-based intervention, asynchronous, synchronous). When it would be possible to summarize, it would be easier to understand and make the tables shorter. Specifically, in the later tables, the hierarchical structure of labeling is difficult to understand, to connect the entries with the correct label category and hierarchy. Some minor problems: Consistency in labeling/naming - e.g. table 4 and sections 3.4.1. and following. Some of the entries are in this form.  Keyword(s): Explanation, others have different patterns of labeling. The font is mixed in the abstract, the form of the abstract is too close to the structure of the paper (in my opinion it is more a table of contents in its current form). Some typos throughout the paper: e.g. line 134 (table legend) IG: intervention group.

The approach presented in the paper is good, the topic is relevant, but it would make sense to revise the structure. 

Author Response

The topic of the paper is of interest, and the work / the efforts the authors have invested in are well documented. However, there are some weaknesses in the presentation. Specifically, the tables (3, 5, 6) are difficult to read and understand.  In some of the cases, tables cover more than one page, but labels are not shown on all pages. In regard to the tables' contents, I think that the details of selected studies could have been condensed to a higher extend. Some information is given repeatedly in each cell (e.g. table 3, telephone intervention, web-based intervention, asynchronous, synchronous). When it would be possible to summarize, it would be easier to understand and make the tables shorter. Specifically, in the later tables, the hierarchical structure of labeling is difficult to understand, to connect the entries with the correct label category and hierarchy.

Response: As suggested, we added the labels of the table for each page. In addition, the details of selected studies were condensed in relation to the tables' contents, and the overall table was summarized in a shorter way by organizing the repeated information in each cell.

Some minor problems: Consistency in labeling/naming - e.g. table 4 and sections 3.4.1. and following.

Response: As suggested, we corrected the labels/names of each table and its section to be consistent throughout the paper.

Keyword(s): Explanation, others have different patterns of labeling. The font is mixed in the abstract, the form of the abstract is too close to the structure of the paper (in my opinion it is more a table of contents in its current form).

Response: We rearranged the keywords and corrected the font of abstract. In addition, the abstract was reorganized focusing on the main results.

Some typos throughout the paper: e.g. line 134 (table legend) IG: intervention group.

Response: We have checked and corrected typos throughout the text, including those you have mentioned.

The approach presented in the paper is good, the topic is relevant, but it would make sense to revise the structure.

Response: Thank you for your opinion on this review. We have tried to revise as you suggested. Please let us know if you have any additional comments on the revision.

Reviewer 2 Report

This article does not add any new information to the already available literature about tele health interventions for self-management in stroke patients. This paper seems more of a compilation of articles rather than a well thought of systematic review. The available information from included papers have been presented in a concise manner but no clear conclusion has been drawn. Above all, the number of studies included are low, not enough to be able to provide any consistent results. One of the conclusion “there are various methods of tele health delivery for self-management” is redundant. This is a well-known fact which the authors have mentioned as their own study conclusion. 

Author Response

Response: Thank you for your comment on this review. We reviewed the literature selection process again, and added one study on self-monitoring. Accordingly, the results and part of the discussion were revised. As you mentioned, although the number of included studies was small and could not be provided consistent results due to the various interventions of TH-SM, the significance of this review was that various TH-SM approaches and contents have been explored in stroke survivors. This point was presented as a limitation in the review. In addition, we have further described. Based on the variety of options for TH-SM support interventions and its applicability in stroke survivors, various trials are needed to establish a consistent basis for the effectiveness of TH-SM intervention in stroke survivors in the future. 

Thank you once more for your comments. Please let us know if there are any additional modifications to this 1st revision.

Round 2

Reviewer 1 Report

The change requests from the first review round were considered, however tables are still complex. But I would now accept a publication. 

Reviewer 2 Report

No further comments.